# Long-COVID is associated with increased absenteeism from work

**Jaewhan Kim**[1]*, **Sanghoon Lee**[2], **Peter Weir**[3]

**1** Department of Physical Therapy, University of Utah, Salt Lake, Utah, United States of America, **2** Department of Economics, Hannam University, Daejeon, South Korea, **3** Medical Group Population Health, University of Utah, Salt Lake, Utah, United States of America

* Jaewhan.kim@utah.edu

## Abstract

Long-COVID, defined as COVID-19 symptoms persisting for more than 3 months, may lead to persistent health issues requiring extensive medical care. Despite its long-term health impact, the economic impact of long-COVID remains understudied. This study examined whether individuals with long-COVID had more missed workdays compared to those without long-COVID. Adults (≥18 years old) with full-time jobs were identified from the 2022 Full-Year Population Characteristics file of the Medical Expenditure Panel Survey (MEPS). A weighted two-part model was used to identify factors associated with missed workdays due to illness. The total population analyzed included 131,685,516 adults (unweighted n = 8,210), with an average (SD) age of 43 (14) years. Among them, 46% were female and 62% were non-Hispanic White. Approximately 7% of the population experienced long-COVID. Individuals with long-COVID reported an average of 8 days missed from work (SD: 12 days), while those without long-COVID reported an average of 4 days missed (SD: 9 days). The two-part model revealed that individuals with long-COVID had 2.54 more missed workdays compared to those without long-COVID (p < 0.01), after controlling for relevant variables. These results underscore significant productivity losses associated with long-COVID, highlighting the need for policymakers and employers to implement effective strategies to address this condition.

## Introduction

Although the declared public health emergency for Coronavirus Disease 2019 (COVID-19) ended in May 2023, the ongoing impacts of the pandemic, including long-COVID, persist, necessitating an ongoing federal response [1]. Long Coronavirus disease (long-COVID) initially gained recognition as a significant health concern through patient reports, which spurred advocacy groups and prompted responses from healthcare professionals, academia, and government entities worldwide.

**Data availability statement:** The data underlying the results presented in the study are available at the following DOI: https://dx.doi.org/10.6084/m9.figshare.27961674.

**Funding:** The author(s) received no specific funding for this work.

**Competing interests:** The authors have declared that no competing interests exist.

Long-COVID could affect anyone who had a COVID-19 infection, leading to chronic health issues requiring comprehensive medical care. Long-COVID is defined as an infection-associated chronic condition that develops following COVID-19 infection and persists for a minimum of 3 months [2,3]. It manifests as a continuous, relapsing and remitting, or progressive disease affecting one or more organs, and includes a broad spectrum of persistent symptoms and conditions that can persist for weeks, months, or even years following a COVID-19 infection. The most frequent symptoms of long-COVID are post-exertional malaise, fatigue, cognitive difficulties (often referred to as "brain fog"), dizziness, gastrointestinal issues, and heart palpitations [4,5].

Since the beginning of the pandemic, there have been over 140 million COVID-19 cases in the U.S. [1] Estimates of the prevalence of long-COVID vary significantly, with some suggesting that anywhere from 10% to 35% or more of individuals infected with COVID-19 may develop the condition [2,4]. In 2022, 6.4% or 6.9% of U.S. adults reported ever having experienced long-COVID [6,7]. According to the U.S. Census Bureau and National Center for Health Statistics Household Pulse Survey, as of April 30 to May 27, 2024, approximately 18.1 percent of all U.S. adults report having ever experienced long-COVID, while 7.5 percent of all U.S. adults indicate they are currently experiencing long-COVID, which suggests about 60 million people in the U.S. could have been affected by long-COVID [8].

Since long-COVID is a newly emerging set of conditions, there are numerous unanswered research questions. Despite the potential impact of long-COVID on the labor market, there is still insufficient information regarding the impact of long-COVID on the working population [9,10]. The aim of this study was to investigate the association between long-COVID and workplace absenteeism as defined by the number of days missed from work. The findings from this study could provide insights into the potential impact of long-COVID on the labor market.

## Methods

### Data

The 2022 Full-Year Population Characteristics file of the Medical Expenditure Panel Survey (MEPS) was used for this study [11]. Conducted annually since 1996, the MEPS has established its reliability and validity through extensive use in various research studies [12–14]. MEPS gathers data from individuals, families, and healthcare professionals across the U.S., covering demographics, health conditions, healthcare utilization and expenditures, insurance status, access to care, employment status, and family relations. The survey sample represents the civilian non-institutionalized population on a national scale. Typically, data for all occupants of a household are provided by a single household respondent or a group of household members collectively [15,16]. The 2022 survey introduced new questionnaires regarding COVID-19, seeking the duration of COVID symptoms (0–12 months) and persistence of COVID-19 symptoms lasting more than 3 months (suggestive of long-COVID) [11]. Since the MEPS data used in this study are publicly available, Internal Review Board (IRB) review was not required.

## Subjects

Adults aged 18 years and older who were employed full-time were included in the study. Subjects with missing variables were excluded from the analysis.

## Outcomes

The primary outcome was the number of days missed from work due to illness or injury within a calendar year.

## Independent variable

The main independent variable was whether a subject experienced long-COVID symptoms, defined as COVID-19 symptoms persisting for more than 3 months following diagnosis (Yes/No). Another independent variable categorized the impact of the COVID-19 infection on missed workdays into three groups: no COVID-19 infection, history of COVID-19 infection, and long-COVID (symptoms persisting for more than 3 months).

## Covariates

The study accounted for factors potentially associated with the outcome. These factors included demographic characteristics and health-related factors [3,17–19]. Demographic variables included age groups (18–30, 31–40, 41–50, 51–64, and ≥65 years), sex (male/female), race/ethnicity (non-Hispanic White, non-Hispanic Black, Hispanic, non-Hispanic Asian, and non-Hispanic other), type of health insurance (private, Medicaid, Medicare, and uninsured), region (Northeast, Midwest, South, and West), marital status (married, widowed/divorced/separated, and never married), and educational attainment (high school graduate: Yes/No). Health-related variables included hypertension (Yes/No), diabetes (Yes/No), asthma (Yes/No), high cholesterol (Yes/No), cancer history (Yes/No), arthritis (Yes/No), any physical limitations (Yes/No), any cognitive limitations (Yes/No), and current smoking status (Yes/No).

## Statistical approaches

All analyses were weighted to ensure they were representative at the population level [11,15,16]. Summary statistics such as means, standard deviations (SD), and percentages described the characteristics of the study subjects. To compare baseline characteristics between the long-COVID and no long-COVID groups, T-tests were used for continuous variables and Chi-square tests were used for categorical variables. Given that about 48% of subjects reported no missed workdays, a two-part model was employed. This approach accounted for potential differences between those with zero missed workdays and those with positive values. The first part used a probit model, while the second part utilized generalized linear regression with a gamma distribution and log link function. Marginal effects of controlled variables were reported from the combined probit and generalized linear regression models of the two-part approach. Sensitivity analysis focused on individuals aged 18–64 (working ages) only. Additionally, negative binomial regression corroborated findings from the two-part model. A p-value less than 0.05 was defined as statistically significant. Stata 18.0 was used for the analysis.

## Results

Among 260,980,061 adults in 2022 (unweighted n = 18,101), subjects with missing values in missed workdays (weighted n = 104,850,051, unweighted n = 8,403) were excluded. Additionally, individuals with missing values in education, hypertension, cancer, high cholesterol, arthritis, physical limitation, cognitive limitation, and smoking status variables (weighted n = 24,444,494, unweighted n = 1,488) were excluded from the analysis. The final study population included 131,685,516 adults (unweighted n = 8,210).

Approximately 7% of adults experienced long-COVID. The average (SD) age of individuals with long-COVID was 45 (13) years, compared to 43 (14) years for those without long-COVID (p-value = 0.03). A higher proportion of females had

long-COVID (57% with long-COVID vs. 45% without long-COVID, p-value < 0.01). Subjects with health conditions such as hypertension (33% vs. 23%, p-value < 0.01), diabetes (10% vs. 7%, p-value = 0.01), asthma (22% vs. 13%, p-value < 0.01), arthritis (24% vs. 14%, p-value < 0.01), physical limitations (8% vs. 4%, p-value < 0.01), and cognitive limitations (4% vs. 1%, p-value < 0.01) were more likely to have long-COVID. Individuals with long-COVID reported an average of 8 days missed from work (SD: 12 days), while those without long-COVID reported 4 days missed (SD: 9 days) (p-value < 0.01) (Table 1).

The marginal effects from the two-part model indicated that individuals with long-COVID experienced an additional 2.54 days missed from work compared to those without long-COVID (p-value < 0.01). Females had 1.45 more missed workdays than males (p-value < 0.01). Subjects with asthma and arthritis were likely to miss an additional 1.52 days (p-value < 0.01) and 2.11 days (p-value < 0.01), respectively, compared to those without these conditions. Similarly, individuals with physical limitations and cognitive limitations had 2.52 days (p-value < 0.01) and 2.54 days (p-value < 0.01) more missed workdays, respectively (Table 2).

In the two-part model considering three groups (no COVID-19 infection, ever had COVID-19 infection, and had long-COVID), individuals with COVID-19 infection experienced 1.90 more missed workdays (p-value < 0.01), and those with long-COVID experienced 4.05 more missed workdays (p-value < 0.01) compared to those with no COVID-19 infection (Table 3).

Another sensitivity analysis included COVID-19 vaccine status (i.e., received a COVID-19 vaccine or booster shot—Yes/No) in the regressions, as vaccination may reduce the risk of short- and long-COVID symptoms. Among the subjects, 83% without long COVID had received the COVID-19 vaccine, compared to 80% of those with long COVID. The difference in vaccination rates between the two groups was not statistically significant (p = 0.124). Furthermore, the COVID-19 vaccine status variable in the regression analysis was not statistically significant and did not affect the main findings.

## Discussion

Long-COVID could have significant impact on various aspects of the economy such as the labor market, productivity, healthcare expenditures, and overall economic growth. It could present a significant economic challenge that complicates the difficulties societies already face in the aftermath of the pandemic. The findings from this study demonstrate that subjects with long-COVID had statistically higher mean days missed from work, based on a nationally representative, cross-sectional sample.

Absenteeism remains a significant problem for many individuals, compromising their ability to work. About 20% of patients with long-COVID remain absent from work for an extended period, specifically for 6 months or more [20]. Restoring health is essential for returning to work, but the delayed onset of persistent symptoms in long-COVID may go unnoticed by workers, impacting their performance and functional capacity to carry out work activities [10].

Long-COVID could affect a significant number of patients, causing chronic health issues that can result in substantial economic impacts. Beyond the initial phase of illness, long-COVID may present ongoing health challenges that diminish workforce productivity and impose financial hardships on both individuals affected and the broader economy [21–23]. A survey conducted by the Swiss Federal Social Insurance Office has documented a growing percentage of disability insurance claims linked to post-COVID-19 conditions [22,23]. Furthermore, long-COVID amplifies the considerable healthcare expenses linked to the COVID-19 pandemic. The extended medical care needed to address long-COVID symptoms, including specialist visits, diagnostic procedures, and rehabilitation services, leads to increased healthcare costs. This adds strain to healthcare systems, diverting resources that could otherwise be allocated to other important areas like preventive healthcare or addressing other health issues [24].

One of the main economic impacts of long-COVID relates to its influence on the labor market. During the COVID-19 pandemic, many conditions such as asthma, arthritis, emotional/nervous/psychiatric problems, vascular/pulmonary/liver conditions, and epilepsy were linked to reduced employment or decreased hours worked, whereas this association was

**Table 1. Characteristics of subjects with and without long-COVID.**

| | Long-COVID | | | |
| | No | Yes | Total | p-value |
| | Mean (sd)/% | Mean (sd)/% | Mean (sd)/% | |
|---|---|---|---|---|
| N (weighted) | 122,227,065 (92.8%) | 9,458,452 (7.2%) | 131,685,516 (100.0%) | |
| Age (continuous) | 43.191 (13.628) | 44.809 (13.134) | 43.307 (13.599) | 0.027 |
| Age category | | | | 0.214 |
| 18-30 | 25,476,171 (20.8%) | 1,735,418 (18.3%) | 27,211,588 (20.7%) | |
| 31-40 | 30,367,219 (24.8%) | 2,185,697 (23.1%) | 32,552,917 (24.7%) | |
| 41-50 | 27,434,948 (22.4%) | 2,010,847 (21.3%) | 29,445,796 (22.4%) | |
| 51-64 | 31,323,223 (25.6%) | 2,931,661 (31.0%) | 34,254,885 (26.0%) | |
| ≥65 | 7,625,503 (6.2%) | 594,828 (6.3%) | 8,220,331 (6.2%) | |
| Sex | | | | <0.001 |
| Male | 67,175,118 (55.0%) | 4,087,927 (43.2%) | 71,263,044 (54.1%) | |
| Female | 55,051,947 (45.0%) | 5,370,525 (56.8%) | 60,422,472 (45.9%) | |
| Race/Ethnicity | | | | <0.001 |
| Non-Hispanic White | 75,343,918 (61.6%) | 6,770,912 (71.6%) | 82,114,830 (62.4%) | |
| Hispanic | 20,834,061 (17.0%) | 1,573,872 (16.6%) | 22,407,933 (17.0%) | |
| Non-Hispanic Black | 13,773,599 (11.3%) | 541,142 (5.7%) | 14,314,741 (10.9%) | |
| Non-Hispanic Asian | 8,567,041 (7.0%) | 319,407 (3.4%) | 8,886,448 (6.7%) | |
| Non-Hispanic others | 3,708,446 (3.0%) | 253,119 (2.7%) | 3,961,565 (3.0%) | |
| Region | | | | 0.629 |
| Northeast | 21,249,277 (17.4%) | 1,654,303 (17.5%) | 22,903,580 (17.4%) | |
| Midwest | 25,987,659 (21.3%) | 2,232,846 (23.6%) | 28,220,505 (21.4%) | |
| South | 46,012,700 (37.6%) | 3,552,599 (37.6%) | 49,565,298 (37.6%) | |
| West | 28,977,429 (23.7%) | 2,018,704 (21.3%) | 30,996,133 (23.5%) | |
| Type of health insurance | | | | 0.815 |
| Private insurance | 94,016,938 (76.9%) | 7,427,078 (78.5%) | 101,444,016 (77.0%) | |
| Medicaid | 11,093,251 (9.1%) | 759,064 (8.0%) | 11,852,314 (9.0%) | |
| Medicare | 7,077,909 (5.8%) | 562,423 (5.9%) | 7,640,331 (5.8%) | |
| No insurance | 10,038,967 (8.2%) | 709,887 (7.5%) | 10,748,855 (8.2%) | |
| Marital status | | | | 0.036 |
| Married | 67,191,352 (55.0%) | 5,364,243 (56.7%) | 72,555,595 (55.1%) | |
| Widowed/divorced/separated | 17,374,230 (14.2%) | 1,659,249 (17.5%) | 19,033,479 (14.5%) | |
| Never married | 37,661,483 (30.8%) | 2,434,960 (25.7%) | 40,096,443 (30.4%) | |
| High school graduate | | | | 0.934 |
| No | 11,465,835 (9.4%) | 898,797 (9.5%) | 12,364,631 (9.4%) | |
| Yes | 110,761,230 (90.6%) | 8,559,655 (90.5%) | 119,320,885 (90.6%) | |
| Hypertension | | | | <0.001 |
| No | 93,801,915 (76.7%) | 6,343,495 (67.1%) | 100,145,411 (76.0%) | |
| Yes | 28,425,149 (23.3%) | 3,114,956 (32.9%) | 31,540,105 (24.0%) | |
| Diabetes | | | | 0.014 |
| No | 113,609,758 (92.9%) | 8,499,905 (89.9%) | 122,109,662 (92.7%) | |
| Yes | 8,617,307 (7.1%) | 958,547 (10.1%) | 9,575,854 (7.3%) | |
| Asthma | | | | <0.001 |
| No | 106,192,298 (86.9%) | 7,336,916 (77.6%) | 113,529,215 (86.2%) | |
| Yes | 16,034,766 (13.1%) | 2,121,536 (22.4%) | 18,156,302 (13.8%) | |

*(Continued)*

**Table 1.** (Continued)

| | Long-COVID | | | |
|---|---|---|---|---|
| | No | Yes | Total | p-value |
| High cholesterol | | | | 0.003 |
| No | 94,994,365 (77.7%) | 6,776,108 (71.6%) | 101,770,473 (77.3%) | |
| Yes | 27,232,699 (22.3%) | 2,682,344 (28.4%) | 29,915,043 (22.7%) | |
| Cancer | | | | 0.721 |
| No | 114,861,489 (94.0%) | 8,853,269 (93.6%) | 123,714,758 (93.9%) | |
| Yes | 7,365,576 (6.0%) | 605,182 (6.4%) | 7,970,758 (6.1%) | |
| Arthritis | | | | <0.001 |
| No | 104,876,801 (85.8%) | 7,210,749 (76.2%) | 112,087,550 (85.1%) | |
| Yes | 17,350,263 (14.2%) | 2,247,703 (23.8%) | 19,597,966 (14.9%) | |
| Physical limitation | | | | <0.001 |
| No | 117,524,853 (96.2%) | 8,666,318 (91.6%) | 126,191,171 (95.8%) | |
| Yes | 4,702,211 (3.8%) | 792,134 (8.4%) | 5,494,346 (4.2%) | |
| Cognitive limitation | | | | <0.001 |
| No | 120,552,443 (98.6%) | 9,111,091 (96.3%) | 129,663,533 (98.5%) | |
| Yes | 1,674,622 (1.4%) | 347,361 (3.7%) | 2,021,983 (1.5%) | |
| Current smoker | | | | 0.146 |
| No | 108,672,261 (88.9%) | 8,183,608 (86.5%) | 116,855,868 (88.7%) | |
| Yes | 13,554,804 (11.1%) | 1,274,844 (13.5%) | 14,829,648 (11.3%) | |
| Number of missed work days | 4.058 (9.208) | 8.057 (12.211) | 4.345 (9.511) | <0.001 |

not observed pre-pandemic [25]. An observational study evaluating patients with long- COVID found significant impacts: more than 54.6% of people experienced prolonged periods of incapacity to work, 34.5% lost their jobs, 63.9% reported difficulties in performing daily activities, and 17.6% faced financial hardships [26]. Another study showed that six months after infection, 56% of patients encountered challenges in daily activities, and 47% could not return to work [27].

Individuals with long-COVID frequently faced difficulties in resuming work or maintaining their previous levels of productivity. Many of them may require extended sick leave or workplace adjustments, resulting in reduced work hours, job loss, or diminished earning potential. As a result, households may suffer income reductions, leading to increasing dependence on social welfare programs, placing additional strain on public resources [24,28]. Nittas et al. reported that between 9% and 40% of patients previously hospitalized due to COVID-19 infection failed to return to work two to three months after their hospitalization, and between 12% and 23% of non-hospitalized patients with mild to moderate symptoms remained absent from work between three and seven months after the acute phase of the disease [29]. Overall, individuals affected by mild COVID or long-COVID often had to reduce or adjust their workload, and some experienced job loss [29]. The impact of long-COVID on the workforce is notably pronounced in service industries such as healthcare, social care, and retail. The labor shortages in these sectors have led to increased wages and prices, contributing to the recent inflationary pressures observed in the U.S. [28]

The demographic characteristics related to long-COVID are noteworthy. While long-COVID can develop in anyone who gets COVID-19 infection, research reported that certain demographic groups are more prone to experiencing it than others, including women, Hispanics, individuals without 4-year college degrees, individuals who have experienced more severe COVID-19 illness, those with underlying health conditions, adults aged 65 or older, and individuals who did not receive a COVID-19 vaccine [3,17–19,30,31]. The age group with the highest percentage of diagnoses typically occurs

**Table 2. Number of workdays missed by subjects with and without long-COVID.**

| Variable | Coefficient | P-value | 95% Confidence Interval | |
|---|---|---|---|---|
| Long-COVID | 2.54 | <0.01 | 1.77 | 3.32 |
| Age category | | | | |
| 18-30 | Reference | | | |
| 31-40 | −0.19 | 0.66 | −1.04 | 0.66 |
| 41-50 | −0.78 | 0.07 | −1.62 | 0.06 |
| 51-64 | −1.06 | 0.02 | −1.93 | −0.20 |
| ≥65 | 2.40 | 0.19 | −1.21 | 6.01 |
| Female | 1.45 | <0.01 | 1.03 | 1.87 |
| Race/Ethnicity | | | | |
| Non-Hispanic White | Reference | | | |
| Hispanic | −0.69 | 0.04 | −1.34 | −0.04 |
| Non-Hispanic Black | −0.48 | 0.25 | −1.31 | 0.34 |
| Non-Hispanic Asian | −1.20 | <0.01 | −1.95 | −0.46 |
| Non-Hispanic others | −0.10 | 0.86 | −1.16 | 0.97 |
| Region | | | | |
| Northeast | Reference | | | |
| Midwest | −0.10 | 0.78 | −0.80 | 0.60 |
| South | 0.19 | 0.60 | −0.53 | 0.92 |
| West | 0.80 | 0.03 | 0.09 | 1.51 |
| Type of health insurance | | | | |
| Private insurance | Reference | | | |
| Medicaid | 0.76 | 0.11 | −0.17 | 1.69 |
| Medicare | −2.90 | <0.01 | −4.09 | −1.71 |
| No insurance | −0.84 | 0.06 | −1.69 | 0.02 |
| Marital status | | | | |
| Married | Reference | | | |
| Widowed/divorced/separated | 0.57 | 0.10 | −0.10 | 1.24 |
| Never married | −0.27 | 0.35 | −0.84 | 0.30 |
| High school graduate | −0.68 | 0.12 | −1.53 | 0.18 |
| Hypertension | 0.47 | 0.13 | −0.14 | 1.08 |
| Diabetes | 0.47 | 0.31 | −0.44 | 1.38 |
| Asthma | 1.52 | <0.01 | 0.73 | 2.31 |
| High cholesterol | 0.39 | 0.13 | −0.12 | 0.90 |
| Cancer | 0.90 | 0.08 | −0.11 | 1.91 |
| Arthritis | 2.11 | <0.01 | 1.40 | 2.82 |
| Physical limitation | 2.52 | <0.01 | 1.47 | 3.56 |
| Cognitive limitation | 2.54 | <0.01 | 0.95 | 4.12 |
| Current smoker | 0.66 | 0.12 | −0.16 | 1.48 |

between the ages of 36 and 50 years, and most long-COVID cases are found in individuals who were not hospitalized and experienced a mild acute illness, reflecting how this population constitutes the majority of overall COVID-19 cases [4].

The findings of this study are consistent with the existing research regarding demographic characteristics. This study showed that the average age of individuals with long-COVID was 45 years, a higher proportion of females had long-COVID, subjects with health conditions such as hypertension, diabetes, asthma, arthritis, physical limitations, and

**Table 3. Impact of COVID-19 on missed workdays: Comparison between no infection, history of COVID-19 infection, and long-COVID.**

| Variable | Coefficient | P-value | 95% Confidence Interval | |
|---|---|---|---|---|
| COVID-19 infection status | | | | |
| No infection | Reference | | | |
| History of COVID-19 infection | 1.90 | <0.01 | 1.40 | 2.40 |
| Long-COVID | 4.05 | <0.01 | 2.88 | 5.22 |
| Age category | | | | |
| 18-30 | Reference | | | |
| 31-40 | −0.20 | 0.64 | −1.05 | 0.65 |
| 41-50 | −0.74 | 0.08 | −1.56 | 0.08 |
| 51-64 | −0.89 | 0.04 | −1.75 | −0.03 |
| ≥65 | 2.32 | 0.19 | −1.16 | 5.81 |
| Female | 1.40 | <0.01 | 0.99 | 1.82 |
| Race/Ethnicity | | | | |
| Non-Hispanic White | Reference | | | |
| Hispanic | −0.66 | 0.05 | −1.31 | 0.00 |
| Non-Hispanic Black | −0.23 | 0.60 | −1.10 | 0.64 |
| Non-Hispanic Asian | −1.08 | 0.01 | −1.84 | −0.32 |
| Non-Hispanic others | −0.04 | 0.93 | −1.08 | 0.99 |
| Region | | | | |
| Northeast | Reference | | | |
| Midwest | −0.08 | 0.83 | −0.78 | 0.62 |
| South | 0.31 | 0.40 | −0.42 | 1.05 |
| West | 0.82 | 0.02 | 0.11 | 1.52 |
| Type of health insurance | | | | |
| Private insurance | Reference | | | |
| Medicaid | 0.85 | 0.07 | −0.06 | 1.77 |
| Medicare | −2.63 | <0.01 | −3.87 | −1.39 |
| No insurance | −0.57 | 0.20 | −1.44 | 0.31 |
| Marital status | | | | |
| Married | Reference | | | |
| Widowed/divorced/separated | 0.63 | 0.06 | −0.03 | 1.30 |
| Never married | −0.18 | 0.54 | −0.74 | 0.39 |
| High school graduate | −0.78 | 0.08 | −1.67 | 0.11 |
| Hypertension | 0.50 | 0.12 | −0.13 | 1.12 |
| Diabetes | 0.45 | 0.32 | −0.45 | 1.35 |
| Asthma | 1.44 | <0.01 | 0.67 | 2.22 |
| High cholesterol | 0.36 | 0.16 | −0.15 | 0.87 |
| Cancer | 0.84 | 0.10 | −0.16 | 1.84 |
| Arthritis | 2.03 | <0.01 | 1.31 | 2.75 |
| Physical limitation | 2.51 | <0.01 | 1.44 | 3.58 |
| Cognitive limitation | 2.54 | <0.01 | 0.93 | 4.14 |
| Current smoker | 0.83 | 0.06 | −0.03 | 1.68 |

A sensitivity analysis restricted to individuals aged 18–64 years yielded similar results. Those with long-COVID experienced 2.28 more missed workdays (p-value < 0.01) compared to those without long-COVID. Negative binomial regression, which was also used with missed workdays as the outcome, corroborated these findings without significant change.

cognitive limitations were more likely to have long-COVID. Moreover, females had 1.44 more missed workdays than males, and individuals with asthma, arthritis, physical limitations, and cognitive limitations were likely to miss more days compared to those without these conditions. These demographic trends could prompt targeted interventions to manage subgroups more likely to be affected by long-COVID.

Considering about 7% (9.2 million) of the population with long-COVID in this study had about 2.5 more missed workdays, the total work loss can be estimated to be 23 million days in a year. If one calculates the daily costs for days of absence based on the average daily earnings as of May 2024 (approximately $279 according to the U.S. Bureau of Labor Statistics [32]), the estimated additional workplace productivity loss costs could be around $6.4 billion due to long-COVID. If more than one year and entire working subjects were considered, productivity loss costs could even bigger. This additional social burden is comparable to the overall cost of obesity to society. Compared to individuals with normal weight, obesity increases job absenteeism by 3 workdays, rising from 2.34 to 5.34 days at the national level. The annual productivity loss due to obesity ranges from $271 to $542 per employee with obesity, with national productivity losses estimated to range from $13.4 to $26.8 billion in 2016 [33].

Gandjour projected the economic, healthcare, and pension costs attributed to long-COVID in Germany in 2021 [34]. The analysis estimated a production loss of 3.4 billion euros and a gross value-added loss of 5.7 billion euros. Additionally, the estimated financial burden on the healthcare and pension systems was about 1.7 billion euros. It implies that the costs associated with long-COVID were significant for the German economy, as well as for its healthcare and pension systems. Comparing to the case of Germany, the productivity loss in the U.S. due to long-COVID is very significant. Moreover, early estimates, factoring in increased healthcare expenses, reduced earnings, and diminished community involvement, suggest that long-COVID could potentially cost the U.S. up to $3.7 trillion [35].

The significant proportion of work absenteeism linked to long-COVID underscores its importance to public health and the economy, and thus provides economic rationale for increased investment in long-COVID treatment, both for employers and policymakers. The ability of people experiencing long-COVID to work could depend on job accommodations and support. Qualitative findings suggest a need for workplace accommodations that are tailored to accommodate fluctuating symptoms, which should be continuously re-evaluated in collaboration between workers and supervisors [36]. Remote work can serve as a practical arrangement for individuals facing activity limitations due to long-COVID. Employers could also consider additional accommodations such as shorter workdays and flexible scheduling to attract and retain employees who are struggling with the effects of long-COVID [17,37].

Social systems assisting those with long-COVID are also important for mitigating social and economic burden. A systemic review with 11 studies from Gualano et al. reported data on individuals experiencing difficulty returning to work after hospitalization or intensive care due to COVID-19 infection [38]. The study highlighted significant differences based on the age and country of origin of the individuals. For example, patients from China and U.S. returned to work more quickly compared to European patients. The authors suggested that these variations could be linked to the age demographics of the patients included in the studies and the differences in social security systems across countries. Additionally, long-COVID patients often encounter skeptical reactions from employers and colleagues, as well as a lack of support from the social welfare system to aid in their return to work [39]. To address the long-COVID challenge, policymakers should consider expanding healthcare coverage, disability benefits, and rehabilitation programs. Given the higher number of missed workdays, governments may need to implement or enhance paid sick leave policies to support affected employees, reducing financial strain and workplace disruptions.

Despite the significant findings in the study, there are several limitations that may affect the findings of this study. First, during the survey years (2021–2022), some subjects may have worked remotely while others worked in-person. This distinction could impact missed workdays, as remote work might be associated with fewer missed days. However, the survey data did not specify the work arrangement (remote vs. in-person). Second, the study relied on self-reported responses, introducing the potential for recall bias which could influence the findings. Third, non-response bias may have occurred due to individuals who did not respond to the survey.

This study used a national survey data to find that individuals with long-COVID experienced additional days missed from work compared to those without long-COVID. The results showed the significant productivity loss caused by missed workdays due to long-COVID, providing an economic justification for policymakers and employers to provide support systems for long-COVID patients to return to work.

## Author contributions

**Conceptualization:** Jaewhan Kim, Sanghoon Lee, Peter Weir.

**Data curation:** Jaewhan Kim.

**Formal analysis:** Jaewhan Kim.

**Investigation:** Sanghoon Lee, Peter Weir.

**Methodology:** Jaewhan Kim, Sanghoon Lee.

**Validation:** Jaewhan Kim, Sanghoon Lee, Peter Weir.

**Writing – original draft:** Jaewhan Kim, Sanghoon Lee, Peter Weir.

**Writing – review & editing:** Jaewhan Kim, Sanghoon Lee, Peter Weir.

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
