## [Decision Letter · Decision Letter 0]

20 Nov 2024

Dear Dr. Kim,

Thank you for submitting your manuscript to PLOS ONE. After careful consideration, we feel that it has merit but does not fully meet PLOS ONE’s publication criteria as it currently stands. Therefore, we invite you to submit a revised version of the manuscript that addresses the points raised during the review process.

We look forward to receiving your revised manuscript.

Kind regards,

Joyce Addo-Atuah, PhD

Academic Editor

PLOS ONE

Journal Requirements:

3. We note you have included a table to which you do not refer in the text of your manuscript. Please ensure that you refer to Table 1,2 and 3 in your text; if accepted, production will need this reference to link the reader to the Table.

Additional Editor Comments:

A very well written manuscript exploring the economic impact of long Covid due to loss of productivity, using the population data available in MEPS, a representative sample of the U.S. population.

In agreement with the reviewer's observation, an individual with subject matter expertise, it would be interesting to factor in the impact of the Covid vaccine on long Covid since the dataset used was from 2021-2022 at a time when the vaccines were freely available.

If MEPS captured this data and you still have access to the dataset and are willing to factor in the use of COVID vaccines in your analysis, that would further enrich the results and make them more meaningful.

In the absence of the datasets or your lack of time to take on the additional work involved, a statement explaining the absence of any reference to the vaccines in the manuscript would be needed.

Reviewers' comments:

Reviewer's Responses to Questions

**Comments to the Author**

1. Is the manuscript technically sound, and do the data support the conclusions?

Reviewer #1: Yes

2. Has the statistical analysis been performed appropriately and rigorously?

Reviewer #1: Yes

3. Have the authors made all data underlying the findings in their manuscript fully available?

Reviewer #1: Yes

4. Is the manuscript presented in an intelligible fashion and written in standard English?

Reviewer #1: Yes

Reviewer #1: Overall this is a great paper addressing the outcomes of long covid on missed work days and its financial impact. What I did not see in the article was the consideration of covid vaccine history as vaccination is the primary method to prevent short and long covid symptoms. The study timeline listed of 2021 to 2022 would include the time when vaccines were made available. If MEPS has this data it would be important to include. If not available, I have no further comments. Good job.

**Do you want your identity to be public for this peer review?** For information about this choice, including consent withdrawal, please see our Privacy Policy

Reviewer #1: No

---

## [Author Response · Author response to Decision Letter 1]

4 Dec 2024

November 30, 2024

Dear Editor and Reviewers,

We appreciate the opportunity to revise our submitted manuscript, Long-COVID is associated with increased absenteeism from work. The reviewer's comments were highly insightful and have contributed to strengthening the quality of this paper. Below are the specific reviewer’s comments (in bold) followed by our responses. Newly added parts in the manuscript are highlighted in yellow. Thank you again for taking the time to consider our manuscript.

Reviewer #1. Overall this is a great paper addressing the outcomes of long covid on missed work days and its financial impact. What I did not see in the article was the consideration of covid vaccine history as vaccination is the primary method to prevent short and long covid symptoms. The study timeline listed of 2021 to 2022 would include the time when vaccines were made available. If MEPS has this data it would be important to include. If not available, I have no further comments. Good job.

Response: Thank you for the comment. We agree that it is critical to consider whether subjects received the COVID-19 vaccine. Using the following information from the MEPS, we determined the vaccination status of the subjects:

• Ever had a COVID-19 vaccine (Yes/No)

• Ever had a COVID-19 booster shot (Yes/No)

Overall, approximately 83% of the subjects had received the COVID-19 vaccine. Among individuals without long COVID, 83% were vaccinated, compared to 80% of those with long COVID. The difference in vaccination rates between the two groups was not statistically significant (p = 0.124).

Table 1. Percent of subjects with COVID vaccine by long COVID status

Long COVID

No Yes Total p-value

N 122,227,065 (92.8%) 9,458,452 (7.2%) 131,685,516 (100.0%)

% of subjects with COVID vaccine 101,505,619 (83.0%) 7,530,853 (79.6%) 109,036,472 (82.8%) 0.124

We accounted for COVID-19 vaccination status in the regression analyses. Tables 2 and 3 below demonstrate the impact of COVID-19 vaccination, as controlled in the regression models.

Table 2. Number of workdays missed by subjects with and without long-COVID

Coefficient P-value 95% Confidence Interval

Long-COVID 2.53 0.00 1.76 3.30

COVID vaccine -0.08 0.79 -0.68 0.52

Age category

18-30 Reference

31-40 -0.16 0.71 -1.00 0.68

41-50 -0.75 0.08 -1.59 0.09

51-64 -1.05 0.02 -1.92 -0.19

≥65 2.49 0.18 -1.18 6.16

Female 1.44 0.00 1.02 1.86

Race/Ethnicity

Non-Hispanic White Reference

Hispanic -0.70 0.04 -1.34 -0.05

Non-Hispanic Black -0.50 0.23 -1.32 0.32

Non-Hispanic Asian -1.21 0.00 -1.96 -0.47

Non-Hispanic others -0.10 0.86 -1.15 0.96

Region

Northeast Reference

Midwest -0.11 0.76 -0.82 0.60

South 0.17 0.64 -0.56 0.91

West 0.81 0.03 0.08 1.53

Type of health insurance

Private insurance Reference

Medicaid 0.74 0.13 -0.21 1.68

Medicare -2.90 0.00 -4.10 -1.70

No insurance -0.85 0.06 -1.75 0.04

Marital status

Married Reference

Widowed/divorced/separated 0.59 0.09 -0.08 1.26

Never married -0.26 0.37 -0.83 0.31

High school graduate -0.65 0.13 -1.49 0.19

Hypertension 0.47 0.14 -0.15 1.08

Diabetes 0.49 0.29 -0.42 1.40

Asthma 1.51 0.00 0.73 2.30

High cholesterol 0.39 0.14 -0.12 0.89

Cancer 0.89 0.08 -0.11 1.88

Arthritis 2.11 0.00 1.40 2.82

Physical limitation 2.51 0.00 1.46 3.56

Cognitive limitation 2.54 0.00 0.94 4.13

Current smoker 0.66 0.12 -0.16 1.48

The COVID-19 vaccine variable is not statistically significant in the regression analysis (coefficient = -0.08, p = 0.79). Controlling for the COVID-19 vaccine variable does not affect the coefficients of the other control variables in the regression. Similarly, the coefficient of the long COVID variable remains consistent between regressions with and without the COVID-19 vaccine variable. Specifically, in the regression without the vaccine variable, the coefficient for long COVID is 2.54 (p < 0.01), while in the regression with the vaccine variable, it is 2.53 (p < 0.01).

Table 3 below presents the results when accounting for COVID-19 infection status categories (no infection, history of COVID-19 infection, and long COVID). In this analysis, the COVID-19 vaccine variable remains statistically insignificant (coefficient = -0.10, p = 0.74). Adding the vaccine variable does not change the coefficients of the COVID-19 infection status variables. For instance, without the vaccine variable, the coefficient for history of COVID-19 infection is 1.90 (p < 0.01), and for long COVID, it is 4.05 (p < 0.01). When the vaccine variable is included, the coefficients are nearly identical: 1.89 (p < 0.01) for history of COVID-19 infection and 4.02 (p < 0.01) for long COVID.

Table 3. Impact of COVID-19 on missed workdays: Comparison between no infection, history of COVID-19 infection, and long-COVID

Coefficient P-value 95% Confidence Interval

COVID-19 infection status

No infection Reference

History of COVID-19 infection 1.89 0.00 1.39 2.39

Long-COVID 4.02 0.00 2.85 5.18

COVID vaccine -0.10 0.74 -0.70 0.50

Age category

18-30 Reference

31-40 -0.17 0.69 -1.01 0.67

41-50 -0.71 0.09 -1.53 0.11

51-64 -0.88 0.05 -1.74 -0.02

≥65 2.41 0.18 -1.13 5.96

Female 1.40 0.00 0.98 1.81

Race/Ethnicity

Non-Hispanic White Reference

Hispanic -0.66 0.05 -1.32 -0.01

Non-Hispanic Black -0.25 0.57 -1.12 0.62

Non-Hispanic Asian -1.09 0.01 -1.85 -0.33

Non-Hispanic others -0.04 0.93 -1.07 0.99

Region

Northeast Reference

Midwest -0.09 0.80 -0.79 0.61

South 0.29 0.45 -0.46 1.04

West 0.82 0.02 0.11 1.53

Type of health insurance

Private insurance Reference

Medicaid 0.83 0.08 -0.10 1.76

Medicare -2.63 0.00 -3.88 -1.38

No insurance -0.59 0.20 -1.50 0.32

Marital status

Married Reference

Widowed/divorced/separated 0.65 0.06 -0.02 1.32

Never married -0.17 0.57 -0.73 0.40

High school graduate -0.75 0.09 -1.62 0.12

Hypertension 0.49 0.12 -0.13 1.12

Diabetes 0.47 0.31 -0.44 1.38

Asthma 1.44 0.00 0.67 2.21

High cholesterol 0.36 0.17 -0.15 0.87

Cancer 0.83 0.10 -0.16 1.82

Arthritis 2.03 0.00 1.31 2.75

Physical limitation 2.50 0.00 1.43 3.58

Cognitive limitation 2.54 0.00 0.92 4.15

Current smoker 0.83 0.06 -0.03 1.69

Additionally, we included interaction terms between long COVID status (no COVID infection, moderate COVID infection, and severe COVID infection) and COVID-19 vaccine status. However, the interaction variables were all statistically insignificant, and the main results for long COVID status remained unchanged.

Since the inclusion of COVID-19 vaccine status did not alter the results, we propose to retain the original reported results in the manuscript. However, we have added the following information to the Results section:

"Another sensitivity analysis included COVID-19 vaccine status (i.e., received a COVID-19 vaccine or booster shot—Yes/No) in the regressions, as vaccination may reduce the risk of short- and long-COVID symptoms. Among the subjects, 83% without long COVID had received the COVID-19 vaccine, compared to 80% of those with long COVID. The difference in vaccination rates between the two groups was not statistically significant (p = 0.124). Furthermore, the COVID-19 vaccine status variable in the regression analysis was not statistically significant and did not affect the main findings.”

Editor Comments:

A very well written manuscript exploring the economic impact of long Covid due to loss of productivity, using the population data available in MEPS, a representative sample of the U.S. population.

In agreement with the reviewer's observation, an individual with subject matter expertise, it would be interesting to factor in the impact of the Covid vaccine on long Covid since the dataset used was from 2021-2022 at a time when the vaccines were freely available.

If MEPS captured this data and you still have access to the dataset and are willing to factor in the use of COVID vaccines in your analysis, that would further enrich the results and make them more meaningful.

In the absence of the datasets or your lack of time to take on the additional work involved, a statement explaining the absence of any reference to the vaccines in the manuscript would be needed.

Response: Please refer to the responses above. We hope that we have adequately addressed your comments.

Journal Requirements:

Response: We double-checked the journal’s style requirements and confirmed that the manuscript aligns with them.

Response: We removed the following ethics statement from the end of the manuscript:

Ethics statement: Since the MEPS data used in this study are publicly available, Internal Review Board (IRB) review was not required.

3. We note you have included a table to which you do not refer in the text of your manuscript. Please ensure that you refer to Table 1,2 and 3 in your text; if accepted, production will need this reference to link the reader to the Table.

Response: We added Table 1 on page 7, Table 2 on page 9, and Table 3 on page 10 to the text.

Response: We reviewed the references, and all are accurate and complete.

Thank you once again for taking the time to review the manuscript!

---

## [Decision Letter · Decision Letter 1]

20 Feb 2025

Dear Dr. Kim,

Thank you for submitting your manuscript to PLOS ONE. After careful consideration, we feel that it has merit but does not fully meet PLOS ONE’s publication criteria as it currently stands. Therefore, we invite you to submit a revised version of the manuscript that addresses the points raised during the review process.

We look forward to receiving your revised manuscript.

Kind regards,

Alex Jones Flores Cassenote, Ph.D.

Academic Editor

PLOS ONE

Journal Requirements:

Additional Editor Comments:

The authors propose a very interesting analysis of the impact of long COVID on absenteeism, using a two-part model to estimate the difference in the number of missed workdays between individuals with and without the condition. The findings reveal that those affected by long COVID experience a significantly higher number of missed workdays, even after controlling for relevant covariates. These results highlight substantial productivity losses, reinforcing the need for policies and strategies aimed at mitigating these impacts in the workplace. Additionally, the study contributes to the discussion on the importance of workplace adaptations and occupational health investments to minimize the effects of long COVID in the labor market. I have only a few specific questions that should be easily addressed by the authors before a final submission.

# Methods and statistical approach

Sample Weighting – The analysis mentions that data were weighted to be representative of the population.

Is there information on the source and calculation of these weights?

Were factors such as age, sex, and occupation considered in the weighting process?

Two-Part Model – This approach is appropriate for handling the asymmetric distribution of missed workdays, but I have a few questions:

What was the rationale for using a probit model in the first part instead of a logistic model?

Why was a generalized linear regression with a gamma distribution chosen for the second part?

Was any model fit test performed to confirm that the gamma distribution was the best choice?

Negative Binomial Regression for Robustness – This is a useful approach for handling overdispersion in count data, but:

Was a Poisson model tested before opting for the negative binomial regression?

Did the results from the negative binomial regression align with those from the two-part model?

Sensitivity Analysis – The decision to focus on individuals aged 18–64 is reasonable since they represent the working-age population, but:

Was any analysis conducted to assess whether the results change significantly when including older adults (65+) or younger individuals (e.g., 16–17 years old)?

Were alternative definitions of the dependent variable tested (e.g., categorizing the number of missed workdays into ranges)?

Potential Biases – Some sources of bias could affect the findings:

Did the study control for pre-existing comorbidities, which may influence the severity of long COVID and the number of missed workdays?

Was vaccination status considered? Since vaccination reduces the likelihood of developing long COVID, it could have a significant impact on the results.

# Discussion

I believe it is important to include a more robust paragraph in the discussion addressing the implications of the findings for public policies and organizational practices. The study highlights the relevance of long COVID for absenteeism and productivity, but it lacks a deeper reflection on how these results can influence the development of occupational health policies, workplace adaptation strategies, and the need for additional investments to mitigate these impacts. Furthermore, practical recommendations for employers and policymakers could strengthen the applicability of the findings and enhance their impact.

Reviewers' comments:

Reviewer's Responses to Questions

**Comments to the Author**

Reviewer #1: All comments have been addressed

Reviewer #2: All comments have been addressed

2. Is the manuscript technically sound, and do the data support the conclusions?

Reviewer #1: Yes

Reviewer #2: Yes

3. Has the statistical analysis been performed appropriately and rigorously?

Reviewer #1: Yes

Reviewer #2: Yes

4. Have the authors made all data underlying the findings in their manuscript fully available?

Reviewer #1: Yes

Reviewer #2: Yes

5. Is the manuscript presented in an intelligible fashion and written in standard English?

Reviewer #1: Yes

Reviewer #2: Yes

Reviewer #1: (No Response)

Reviewer #2: The study presents the results of original research. Methods are described in sufficient detail and sample size is large enough to produce robust results. The data presented in the manuscript must support the conclusions

**Do you want your identity to be public for this peer review?** For information about this choice, including consent withdrawal, please see our Privacy Policy

Reviewer #1: No

Reviewer #2: No

---

## [Author Response · Author response to Decision Letter 2]

19 Mar 2025

March 19, 2025

Dear Dr. Cassenote and Reviewers,

We appreciate the opportunity to revise our submitted manuscript, Long-COVID is associated with increased absenteeism from work. Below are the specific reviewer’s comments (in bold) followed by our responses. Newly added parts in the manuscript are highlighted in yellow. Thank you again for taking the time to consider our manuscript.

Journal Requirements:

Response: We have reviewed the references, and they are all complete and correct.

Additional Editor Comments:

The authors propose a very interesting analysis of the impact of long COVID on absenteeism, using a two-part model to estimate the difference in the number of missed workdays between individuals with and without the condition. The findings reveal that those affected by long COVID experience a significantly higher number of missed workdays, even after controlling for relevant covariates. These results highlight substantial productivity losses, reinforcing the need for policies and strategies aimed at mitigating these impacts in the workplace. Additionally, the study contributes to the discussion on the importance of workplace adaptations and occupational health investments to minimize the effects of long COVID in the labor market. I have only a few specific questions that should be easily addressed by the authors before a final submission.

Additional Editor Comments:

The authors propose a very interesting analysis of the impact of long COVID on absenteeism, using a two-part model to estimate the difference in the number of missed workdays between individuals with and without the condition. The findings reveal that those affected by long COVID experience a significantly higher number of missed workdays, even after controlling for relevant covariates. These results highlight substantial productivity losses, reinforcing the need for policies and strategies aimed at mitigating these impacts in the workplace. Additionally, the study contributes to the discussion on the importance of workplace adaptations and occupational health investments to minimize the effects of long COVID in the labor market. I have only a few specific questions that should be easily addressed by the authors before a final submission.

# Methods and statistical approach

Sample Weighting – The analysis mentions that data were weighted to be representative of the population.

Is there information on the source and calculation of these weights?

Were factors such as age, sex, and occupation considered in the weighting process?

Response: The detailed methodology for sample weighting can be found on the following website:

11. Agency for Healthcare Research and Quality. MEPS HC 238: 2022 Full-Year Population Characteristics. Accessed June 01, 2024, https://meps.ahrq.gov/data_stats/download_data/pufs/h238/h238doc.shtml

This source was cited in the manuscript because it provides a comprehensive description of the data used. Specifically, Section 3.0: Survey Sample Information details the sample weighting methodology. For example,

“Section 3.1. Sample Weights and Variance Estimation

Weight variables in the 2022 PC PUF can be used to generate estimates of totals, means, percentages, and rates for persons and families in the U.S. civilian noninstitutionalized population. Procedures and considerations associated with the construction and interpretation of person- and family-level estimates using these and other variables are discussed in this section. It should be noted that NCHS has made a modification to the NHIS sample design that has affected the MEPS variance structure. This is discussed in detail in Section 3.6.1.”

Regarding specific weighting factors, demographic characteristics such as age and sex were included in the weighting process. However, occupation was not a factor in the weighting calculations. More detailed information can be found in the source below:

Machlin S.R., Chowdhury S.R., Ezzati-Rice T., DiGaetano R., Goksel H., Wun L.-M., Yu W., Kashihara D. Estimation Procedures for the Medical Expenditure Panel Survey Household Component. Methodology Report #24. September 2010. Agency for Healthcare Research and Quality, Rockville, MD.

Two-Part Model – This approach is appropriate for handling the asymmetric distribution of missed workdays, but I have a few questions:

What was the rationale for using a probit model in the first part instead of a logistic model?

Response: The probit model was chosen due to its strong theoretical connection to normality (i.e., a normal distribution of the latent propensity to miss workdays), which can be advantageous when modeling outcomes influenced by a continuous, normally distributed latent propensity.

Why was a generalized linear regression with a gamma distribution chosen for the second part?

Response: Among participants who missed workdays, the number of missed days was right-skewed, meaning most individuals had relatively few missed days, while a smaller number had very high values. To account for this skewed distribution with only positive values, we selected a generalized linear regression model with a gamma distribution. In addition, we conducted a Modified Park test to determine the best-fitting distribution.

Was any model fit test performed to confirm that the gamma distribution was the best choice?

Response: We tested Gaussian, Poisson, and gamma distribution families with various link functions, including log and identity link functions, to determine the best distribution-link function combination. To select the optimal distribution, we considered the Modified Park test, Akaike Information Criterion (AIC), and Bayesian Information Criterion (BIC). Both the Gaussian family with an identity link function and the gamma family with a log link function performed well; however, the gamma family with a log link function had lower AIC and BIC values. Therefore, we selected the gamma distribution.

Negative Binomial Regression for Robustness – This is a useful approach for handling overdispersion in count data, but:

Was a Poisson model tested before opting for the negative binomial regression?

Response: The variance of the outcome variable is much greater than its mean (variance = 135, mean = 4.345), indicating overdispersion.

We ran and compared Poisson and negative binomial regression models, both of which produced similar results for the independent variables—long COVID (Table 1) and COVID infection status (Table 2).

Table 1. Incidence rate ratio (IRR) of long COVID from Poisson and negative binomial regressions

Poisson regression Negative binomial regression

Variable IRR P-value 95% Confidence Interval IRR P-value 95% Confidence Interval

Long-COVID 1.63 <0.01 1.38 1.93 1.83 <0.01 1.53 2.18

*Note: The other control variables are not presented in the table.

Both regressions generated robust standard errors by incorporating survey methodology.

Table 2. Incidence rate ratio (IRR) of COVID infection status from Poisson and negative binomial regressions

Poisson regression Negative binomial regression

Variable Coefficient P-value 95% Confidence Interval Coefficient P-value 95% Confidence Interval

COVID-19 infection status

No infection Reference Reference

History of COVID-19 infection 1.64 <0.01 1.43 1.89 1.70 <0.01 1.48 1.95

Long-COVID 2.21 <0.01 1.81 2.70 2.52 <0.01 2.04 3.11

*Note: The other control variables are not presented in the table.

Both regressions generated robust standard errors by incorporating survey methodology.

Did the results from the negative binomial regression align with those from the two-part model?

Response: The independent variables and control variables from both models are similar. Tables 3 and 4 below present the marginal effects of the independent variables from the two-part model and the negative binomial model.

Table 3. Comparison of the coefficient of the long COVID variable between the two-part model and the negative binomial model

Two-part model Negative binomial model

Variable Coefficient P-value 95% Confidence Interval Coefficient P-value 95% Confidence Interval

Long-COVID 2.54 <0.01 1.77 3.32 2.67 <0.01 1.86 3.48

*Note: The other control variables are not presented in the table.

Table 4. Comparison of the coefficients of the COVID-19 status variables between the two-part model and the negative binomial model

Two-part model Negative binomial model

Variable Coefficient P-value 95% Confidence Interval Coefficient P-value 95% Confidence Interval

COVID-19 infection status

No infection Reference

History of COVID-19 infection 1.90 <0.01 1.40 2.40 2.09 <0.01 1.58 2.61

Long-COVID 4.05 <0.01 2.88 5.22 4.58 <0.01 3.23 5.92

*Note: The other control variables are not presented in the table.

We have already described the following sentence in the Results section:

“Negative binomial regression, which was also used with missed workdays as the outcome, corroborated these findings without significant change.”

Sensitivity Analysis – The decision to focus on individuals aged 18–64 is reasonable since they represent the working-age population, but:

Was any analysis conducted to assess whether the results change significantly when including older adults (65+) or younger individuals (e.g., 16–17 years old)?

Response: The original analyses included all adults (≥18 years old), including the elderly population (65+). A sensitivity analysis was conducted with subjects aged 18–64 years, as described in the Methods section:

“Sensitivity analysis focused on individuals aged 18-64 (working ages) only.”

The results with and without the elderly population were similar. We have previously described the following in the Results section:

“A sensitivity analysis restricted to individuals aged 18 to 64 years yielded similar results. Those with long-COVID experienced 2.28 more missed workdays (p-value < 0.01) compared to those without long-COVID.”

Responses for children (<18 years old) regarding long COVID are not available in the original dataset due to confidentiality reasons indicated by AHRQ. More detailed information is available on the AHRQ website:

Agency for Healthcare Research and Quality. MEPS HC 238: 2022 Full-Year Population Characteristics. https://meps.ahrq.gov/data_stats/download_data/pufs/h238/h238doc.shtml

Were alternative definitions of the dependent variable tested (e.g., categorizing the number of missed workdays into ranges)?

Response: We did not consider alternative definitions of the dependent variable, such as categorizing the number of missed workdays. If we were to categorize the dependent variable, we would need to provide rationales or justification for doing so. Additionally, we believe that categorizing the dependent variable is unnecessary for analyzing the data and does not offer any benefits compared to treating the dependent variable as continuous.

Potential Biases – Some sources of bias could affect the findings:

Did the study control for pre-existing comorbidities, which may influence the severity of long COVID and the number of missed workdays?

Response: We have already included several pre-existing comorbidities, such as hypertension, diabetes, and asthma. These are described in the Methods section.

“Health-related variables included hypertension (Yes/No), diabetes (Yes/No), asthma (Yes/No), high cholesterol (Yes/No), cancer history (Yes/No), arthritis (Yes/No), any physical limitations (Yes/No), any cognitive limitations (Yes/No), and current smoking status (Yes/No).”

Was vaccination status considered? Since vaccination reduces the likelihood of developing long COVID, it could have a significant impact on the results.

Response: This was considered in Revision 1. Below were our responses in Revision 1.

Using the following information from the MEPS, we determined the vaccination status of the subjects:

• Ever had a COVID-19 vaccine (Yes/No)

• Ever had a COVID-19 booster shot (Yes/No)

Overall, approximately 83% of the subjects had received the COVID-19 vaccine. Among individuals without long COVID, 83% were vaccinated, compared to 80% of those with long COVID. The difference in vaccination rates between the two groups was not statistically significant (p = 0.124).

Table 1. Percent of subjects with COVID vaccine by long COVID status

Long COVID

No Yes Total p-value

N 122,227,065 (92.8%) 9,458,452 (7.2%) 131,685,516 (100.0%)

% of subjects with COVID vaccine 101,505,619 (83.0%) 7,530,853 (79.6%) 109,036,472 (82.8%) 0.124

We accounted for COVID-19 vaccination status in the regression analyses. Tables 2 and 3 below demonstrate the impact of COVID-19 vaccination, as controlled in the regression models.

Table 2. Number of workdays missed by subjects with and without long-COVID

Coefficient P-value 95% Confidence Interval

Long-COVID 2.53 0.00 1.76 3.30

COVID vaccine -0.08 0.79 -0.68 0.52

Age category

18-30 Reference

31-40 -0.16 0.71 -1.00 0.68

41-50 -0.75 0.08 -1.59 0.09

51-64 -1.05 0.02 -1.92 -0.19

≥65 2.49 0.18 -1.18 6.16

Female 1.44 0.00 1.02 1.86

Race/Ethnicity

Non-Hispanic White Reference

Hispanic -0.70 0.04 -1.34 -0.05

Non-Hispanic Black -0.50 0.23 -1.32 0.32

Non-Hispanic Asian -1.21 0.00 -1.96 -0.47

Non-Hispanic others -0.10 0.86 -1.15 0.96

Region

Northeast Reference

Midwest -0.11 0.76 -0.82 0.60

South 0.17 0.64 -0.56 0.91

West 0.81 0.03 0.08 1.53

Type of health insurance

Private insurance Reference

Medicaid 0.74 0.13 -0.21 1.68

Medicare -2.90 0.00 -4.10 -1.70

No insurance -0.85 0.06 -1.75 0.04

Marital status

Married Reference

Widowed/divorced/separated 0.59 0.09 -0.08 1.26

Never married -0.26 0.37 -0.83 0.31

High school graduate -0.65 0.13 -1.49 0.19

Hypertension 0.47 0.14 -0.15 1.08

Diabetes 0.49 0.29 -0.42 1.40

Asthma 1.51 0.00 0.73 2.30

High cholesterol 0.39 0.14 -0.12 0.89

Cancer 0.89 0.08 -0.11 1.88

Arthritis 2.11 0.00 1.40 2.82

Physical limitation 2.51 0.00 1.46 3.56

Cognitive limitation 2.54 0.00 0.94 4.13

Current smoker 0.66 0.12 -0.16 1.48

The COVID-19 vaccine variable is not statistically significant in the regression analysis (coefficient = -0.08, p = 0.79). Controlling for the COVID-19 vaccine variable does not affect the coefficients of the other control variables in the regression. Similarly, the coefficient of the long COVID variable remains consistent between regressions with and without the COVID-19 vaccine variable. Specifically, in the regression without the vaccine variable, the coefficient for long COVID is 2.54 (p < 0.01), while in the regression with the vaccine variable, it is 2.53 (p < 0.01).

Table 3 below presents the results when accounting for COVID-19 infection status categories (no infection, history of COVID-19 infection, and long COVID). In this analysis, the COVID-19 vaccine variable remains statistically insignificant (coefficient = -0.10, p = 0.74). Adding the vaccine variable does not change the coefficients of the COVID-19 infection status variables. For instance, without the vaccine variable, the coefficient for history of COVID-19 infection is 1.90 (p < 0.01), and for long COVID, it is 4.05 (p < 0.01). When the vaccine variable is included, the coefficients are nearly identical: 1.89 (p < 0.01) for history of COVID-19 infection and 4.02 (p < 0.01) for long COVID.

Table 3. Impact of COVID-19 on missed workdays: Comparison between no infection, history of COVID-19 infection, and long-COVID

Coefficient P-value 95% Confidence Interval

COVID-19 infection status

No infection Reference

History of COVID-19 infection 1.89 0.00 1.39 2.39

Long-CO

---

## [Decision Letter · Decision Letter 2]

12 May 2025

Long-COVID is associated with increased absenteeism from work

PONE-D-24-37720R2

Dear Dr. Kim,

We’re pleased to inform you that your manuscript has been judged scientifically suitable for publication and will be formally accepted for publication once it meets all outstanding technical requirements.

Kind regards,

Tae-Young Pak, Ph.D.

Academic Editor

PLOS ONE

Additional Editor Comments (optional):

Reviewers' comments:

Reviewer's Responses to Questions

**Comments to the Author**

Reviewer #2: All comments have been addressed

2. Is the manuscript technically sound, and do the data support the conclusions?

Reviewer #2: Yes

3. Has the statistical analysis been performed appropriately and rigorously?

Reviewer #2: Yes

4. Have the authors made all data underlying the findings in their manuscript fully available?

Reviewer #2: Yes

5. Is the manuscript presented in an intelligible fashion and written in standard English?

Reviewer #2: Yes

Reviewer #2: The study presents the results of original research. Methods were

described in sufficient detail and conclusions are consistent with data. All issues were adressed in revision.

**Do you want your identity to be public for this peer review?** For information about this choice, including consent withdrawal, please see our Privacy Policy

Reviewer #2: No

---

## [Editor Report · Acceptance letter]

PONE-D-24-37720R2

PLOS ONE

Dear Dr. Kim,

I'm pleased to inform you that your manuscript has been deemed suitable for publication in PLOS ONE. Congratulations! Your manuscript is now being handed over to our production team.

Kind regards,

on behalf of

Dr. PLOS Manuscript Reassignment

Staff Editor

PLOS ONE